# Pan-Genome Analysis of *Wolbachia*, Endosymbiont of *Diaphorina citri*, Reveals Independent Origin in Asia and North America

**DOI:** 10.3390/ijms25094851

**Published:** 2024-04-29

**Authors:** Jiahui Zhang, Qian Liu, Liangying Dai, Zhijun Zhang, Yunsheng Wang

**Affiliations:** 1Hunan Provincial Key Laboratory for Biology and Control of Plant Diseases and Insect Pests, College of Plant Protection, Hunan Agricultural University, Changsha 410128, China; 15096100996zjh@gmail.com (J.Z.); qianliu7427@163.com (Q.L.); daily@hunau.net (L.D.); 2State Key Laboratory for Managing Biotic and Chemical Threats to the Quality and Safety of Agro-Products, Institute of Plant Protection and Microbiology, Zhejiang Academy of Agricultural Sciences, Hangzhou 310021, China

**Keywords:** *Wolbachia*, pan-genome, Asian citrus psyllid, metabolic pathways

## Abstract

*Wolbachia*, a group of Gram-negative symbiotic bacteria, infects nematodes and a wide range of arthropods. *Diaphorina citri* Kuwayama, the vector of *Candidatus* Liberibacter asiaticus (*C*Las) that causes citrus greening disease, is naturally infected with *Wolbachia* (*w*Di). However, the interaction between *wDi* and *D. citri* remains poorly understood. In this study, we performed a pan-genome analysis using 65 *w*Di genomes to gain a comprehensive understanding of *w*Di. Based on average nucleotide identity (ANI) analysis, we classified the *w*Di strains into Asia and North America strains. The ANI analysis, principal coordinates analysis (PCoA), and phylogenetic tree analysis supported that the *D. citri* in Florida did not originate from China. Furthermore, we found that a significant number of core genes were associated with metabolic pathways. Pathways such as thiamine metabolism, type I secretion system, biotin transport, and phospholipid transport were highly conserved across all analyzed *w*Di genomes. The variation analysis between Asia and North America *w*Di showed that there were 39,625 single-nucleotide polymorphisms (SNPs), 2153 indels, 10 inversions, 29 translocations, 65 duplications, 10 SV-based insertions, and 4 SV-based deletions. The SV-based insertions and deletions involved genes encoding transposase, phage tail tube protein, ankyrin repeat (ANK) protein, and group II intron-encoded protein. Pan-genome analysis of *w*Di contributes to our understanding of the geographical population of *w*Di, the origin of hosts of *D. citri*, and the interaction between *w*Di and its host, thus facilitating the development of strategies to control the insects and huanglongbing (HLB).

## 1. Introduction

*Wolbachia*, a group of Gram-negative bacteria, belongs to the group of α-proteobacteria. They are matrilineally inherited endosymbionts and are commonly found in a variety of organisms, including insects, spiders, nematodes, mites, and plant nematodes [1]. *Wolbachia* has the remarkable ability to manipulate the reproductive phenotypes of its arthropod hosts. This includes inducing cytoplasmic incompatibility, which is the most common reproductive phenotype in arthropods, as well as inducing parthenogenesis, feminization, and male killing [2]. There exist facultative and obligate mutualisms between *Wolbachia* and their hosts. In the cases of facultative mutualism, typically associated with arthropod hosts, *Wolbachia* can provide a positive fecundity advantage. For instance, in *Drosophila melanogaster* reared on iron-restricted or iron-overloaded diets, *Wolbachia* contributes to increased fecundity [3]. In the context of obligate mutualism, both *Wolbachia* and their hosts benefit. When *Wolbachia* is removed via tetracycline treatment, it hinders the development, growth, and fecundity of host nematodes. In contrast, *Wolbachia*-free hosts remain unaffected [4]. In the case of *Brugia malayi*, *Wolbachia* lacks complete pathways for the de novo biosynthesis of essential molecules such as Coenzyme A, NAD, biotin, lipoic acid, ubiquinone, folate, and pyridoxal phosphate. This suggests that *Wolbachia* relies on its host to supply these essential compounds. Moreover, *Wolbachia* in *B. malay* has the ability to provide riboflavin, flavin adenine dinucleotide, heme, and nucleotides to the host [5]. The presence of *Wolbachia* in insects has been shown to suppress viral replication. A notable example is the use of *Wolbachia* to block dengue, Chikungunya, and Zika virus [6]. In the case of plant viruses, such as ragged rice stunt virus (RRSV), the presence of *Wolbachia* can inhibit RRSV infection and transmission while mitigating virus-induced symptoms in rice plants [7].

The Asian citrus psyllid (ACP), *Diaphorina citri* Kuwayama (Hemiptera: Psyllidae), boasts a wide geographic distribution, with its likely origin traced back to Asia [8]. It was initially described in Taiwan in 1907 [9]. Over time, *D. citri* has achieved a global presence and is currently found in various parts of the world. It has established a significant presence throughout Asia, and in the 1940s, it was found in Brazil [10]. From there, it spread to Florida [11] and has since infested most of the citrus-producing states in the United States [12,13]. The Caribbean and the southern regions of the United States have also reported the presence of this psyllid [8,14]. *D. citri* serves as the insect vector responsible for transmitting *C*Las, a pathogen that triggers immune responses in the phloem tissue of citrus, ultimately resulting in citrus greening disease, also known as HLB. Like many insects, all developmental stages of *D. citri* are also infected with *Wolbachia* [15]. Recent studies indicate that the presence of *C*Las may increase the proportion of *Wolbachia* when compared to uninfected conditions in ACP [16]. Furthermore, *w*Di may play a role in regulating the phage lytic cycle genes in *C*Las [17]. Unfortunately, currently, there is no known cure for HLB-infected plants [18,19]. The potential interactions between *Wolbachia* and its ACP host may hold promise for the development of novel strategies for the treatment of ACP and HLB.

The intimate interaction between a parasite and its host, coupled with the extensive vertical transmission of the symbiont across host generations, strongly implies a shared evolutionary history between the parasite and its host [20]. *Candidatus* Carsonella ruddii serves as the primary endosymbiont in *D. citri*. Existing research indicates a notable phylogenetic congruence between *D. citri* and *C. ruddii*. Moreover, genetic markers of the endosymbiont hold promise for exploring the evolutionary history and biogeographical patterns of *D. citri* [21]. Considering the potential interactions between *w*Di and the host, along with the vertical transmission of *w*Di, it becomes plausible to utilize *w*Di as a proxy for tracing the evolutionary history of its host. Previously, Saha et al. [22] showed that *w*Di in Floridian *D. citri* belonged to a sub-clade of supergroup B different from *w*Di in Chinese *D. citri* and suggested that *D. citri* in Florida did not originate from China.

*D. citri* is a widespread insect that causes significant losses in the global citrus industry due to the transmission of HLB. Considering the application of *Wolbachia* in blocking viruses, *w*Di has the potential to be used to control *D. citri* or *D. citri*-borne pathogens. In this study, we conduct a pan-genome analysis of *w*Di that may facilitate the understanding of *w*Di–*D. citri* interactions and benefit the development of control strategies for *D. citri* and HLB in different regions.

## 2. Results

### 2.1. The Genome Characteristics

The genome data for 13 *w*Di strains (GCA_000331595, GCA_013096355, GCA_013096535, GCA_013096725, GCA_013458815, GCA_017883655, GCA_017883735, GCA_017883805, GCA_017883845, GCA_017883905, GCA_019355235, GCA_019355355, and GCA_019355375) are available in the National Center for Biotechnology Information (NCBI) databases under the following BioProject Accessions: PRJNA29451, PRJNA603775, PRJNA544530, PRJNA704462, and PRJNA603775. We assembled an additional 52 *w*Di genomes, and the raw read data for these genomes can be found in the NCBI Sequence Read Archive (SRA) database (Appendix A). Among these, two *w*Di genomes were assembled from the sequencing datasets of Taiwan *D. citri*, while only one *w*Di genome was obtained from the sequencing datasets of California and Uruguay *D. citri*. These 52 *w*Di genomes consist of 1 complete circular genome and 51 genomes at the contig level, with genome sizes ranging from 1,240,904 to 2,318,766 bp. The BUSCO completeness scores for these genomes vary from 71.70% to 99.80%. The *w*Di genome *w*DiTW_1 exhibits lower BUSCO completeness scores, 71.70%, in contrast to the approximately 95.00% observed in the majority of other *w*Di genomes (Appendix A). The number of ANK proteins in the 65 *w*Di genomes ranges from 37 to 124.

The five multilocus sequence typing (MLST) alleles, *coxA*, *fbpA*, *ftsZ*, *gatB*, and *hcpA*, in the six *w*Di strains in Guilin, China, and *w*DiUY in Uruguay, completely matched ST-173 in the PubMLST database [23]. The profile of the 56 *w*Di strains in the USA was referred to as ST-FL. These two prevalent *w*Di strains, ST-173 and ST-FL, were identified previously [24]. It is worth noting that *w*DiTW_1 possesses two copies of *coxA* and *fbpA* genes, along with a *ftsZ* gene, which are part of the ST-173 profile while lacking *gatB* and *hcpA*. Additionally, the MLST profile of *w*DiTW_2 corresponds to ST-462 in the PubMLST database [23].

### 2.2. Identification of Two wDi Types Based on ANI Analysis, PCoA, and Phylogenetic Analysis

Genetic divergence was assessed by evaluating the 65 *w*Di overall nucleotide sequence using fastANI v1.33 [25]. The ANI analysis result revealed a range of ANI values, spanning from 95.00417 to 99.99884, with the exclusion of self-comparisons in the results (Appendix A). Specifically, when comparing 56 *w*Di strains within the USA, the ANI ranged from 98.32615 to 99.99884. In the case of *w*Di strains from Guilin, the ANI ranged from 99.64244 to 99.87062. The ANI value between Taiwan *w*DiTW_1 and Uruguay *w*DiUY was exceptionally high at 99.69395, and their ANI value with Guilin *w*Di was also quite high. The ANI value comparing *w*DiTW_2 with 56 USA *w*Di was approximately 97, while the ANI value between *w*DiTW_2 and *w*DiTW_1, *w*DiUY, and *w*Di strains from Guilin was around 96. The ANI clustering segregated the 65 *w*Di into two distinct categories (Figure 1). The USA *w*Di strains formed one cluster, while Guilin, Taiwan, and Uruguay *w*Di strains clustered together. As a result, we categorized them as North America *w*Di (NA_*w*Di) and Asia *w*Di (AS_*w*Di) strains, respectively. We detected gene family analysis across all 65 *w*Di genomes using OrthoFinder v2.5.4 [26]. This analysis revealed that all genes from these genomes were classified into 1916 gene families. Subsequently, we compared gene copy number variations among all these families using PCoA. A comparison of all 65 *w*Di genomes was performed based on 1916 gene families. Despite variations in the number of CDS (ranging from 1088 to 2299; average ± standard deviation: 1329 ± 190; Appendix A), the 56 NA_*w*Di genomes formed a distinct cluster in the PCoA analysis (Figure 2). Additionally, the nine AS_*w*Di formed their separate clusters (Figure 2). The resulting PCoA clustering patterns (Figure 2) were generally consistent with the previously observed ANI clustering (Figure 1), with two clusters. The phylogenetic tree also shows that AS_*w*Di and NA_*w*Di were clustered in different branches, and *w*DiTW_2 did not cluster with the AS_*w*Di (Figure 3).

### 2.3. wDi Possesses a Closed Pan-Genome

To assess whether the pan-genome was open or closed, we generated core genome and pan-genome development plots for *w*Di based on 100 random combinations of the 65 *w*Di genomes (Figure 4A). These plots illustrated that as more genomes were added, the number of genes in the pan-genome approached saturation, while the number of genes in the core genome decreased. The analysis indicated that the pan-genome of *w*Di could be characterized as “closed”, supported by the *B*_pan_ < 0.

For the gene sets, a total of 504 (26.3%) core orthogroups were identified in all 65 *w*Di genomes, and the genes within these core orthogroups were categorized as core genes. Additionally, 490 (25.6%) softcore orthogroups were found in 59 to 64 *w*Di genomes, and the genes in these softcore orthogroups were defined as softcore genes. Another 902 (47.1%) dispensable orthogroups were detected in 2 to 58 *w*Di genomes, categorizing the genes within them as dispensable genes. Lastly, 20 (1%) private orthogroups were unique to a single *w*Di genome each. The genes in these private orthogroups were designated as private genes (Figure 4B). The private genes have also been included in these genes from the non-clustered orthogroups. The presence/absence of all 1916 orthogroups in 65 genomes was visualized by a generated heatmap (Figure 4C). The heatmap revealed that there were 17 orthogroups exclusively found in strains from North America and 5 orthogroups unique to strains from Asia. Furthermore, we conducted multiple comparisons of the lengths of core, softcore, dispensable, and private genes using the Student–Newman–Keuls test with α = 0.05. The results of these multiple comparisons revealed significant differences in the lengths of core, softcore, dispensable, and private genes (Figure 4D). Our analysis revealed that a substantial proportion of the core genes, specifically 81.3%, and softcore genes, totaling 72.0%, contained Pfam domains. These percentages were higher than those observed in the dispensable and private genes, which stood at 45.9% and 31.4%, respectively (Figure 4E). A detailed orthogroup presence and absence matrix, as well as length statistics for all genes, is available in Appendix A.

### 2.4. Core Genes Associated with Metabolism, and SNP Analysis Revealing Potential Co-Infection of wDiTW_2

The Clusters of Orthologous Genes (COG), Gene Ontology (GO), and Kyoto Encyclopedia of Genes and Genomes Orthology (KO) of 37,376 core genes were assigned by using eggNOG-Mapper v2.1.10 [31] against the eggNOG database v5.0.2 [32]; these core genes may play an essential function in *w*Di. The core genes were annotated into four major functional categories: metabolism (37.00%), information storage and processing (22.71%), poorly characterized (20.80%), and cellular processes and signaling (19.49%) (Figure 5A). It is noteworthy that number of core genes annotated as metabolism genes was particularly high, including energy production and conversion (2543 genes), coenzyme transport and metabolism (1790 genes), carbohydrate transport and metabolism (2224 genes), and nucleotide transport and metabolism (2052 genes). The GO annotations indicated that the genes were primarily linked to biological processes, followed by molecular functions and cellular components. Among biological processes, both the cellular processes (679 genes) and metabolic processes (549 genes) emerged as the most significant categories (Figure 5B). Regarding cellular components, 614 core genes were associated with cellular anatomical entity (Figure 5B). In terms of molecular functions, the top three categories were catalytic activity (475 genes), binding (332 genes), and structural molecule activity (130 genes) (Figure 5B). Among Kyoto Encyclopedia of Genes and Genomes (KEGG) annotations, the top three pathways were metabolism (8480 genes), genetic information processing (5020 genes), and environmental information processing (2781 genes) (Figure 5C). Within the metabolism pathways, there was a significant representation of energy metabolism (2510 genes) and metabolism of cofactors and vitamins (2386 genes) (Figure 5C). Furthermore, in genetic information processing, a substantial number of core genes, totaling 2521, were associated with translation (Figure 5C).

The top 8 out of 10 enriched GO terms from the results of the GO enrichment analysis were represented by 203 genes each. These terms were tricarboxylic acid cycle, citrate metabolic process, aerobic respiration, antibiotic metabolic process, tricarboxylic acid metabolic process, energy derivation by oxidation of organic compounds, cellular respiration, and generation of precursor metabolites and energy (Figure 6A). Additionally, protein folding and pyrimidine-containing compound metabolic process involved 135 and 130 genes, respectively (Figure 6A). The KEGG enrichment analysis revealed that the core genes were mainly involved in methane metabolism, ubiquinone and other terpenoid-quinone biosynthesis, pentose phosphate pathway, and protein processing in the endoplasmic reticulum (Figure 6B).

A total of 7675 recombination-free core SNPs, 1 core SNP on average every about 199 bases, were identified through alignment to the reference genome GCA_019355355 using snippy v4.6.0 [33]. Additionally, 21 multi-allelic mutations were detected in the analysis. Among these SNPs, 4348 were identified as synonymous variants, while 2851 were classified as missense variants. A subset of 26 SNPs was predicted to have significant implications for the associated genes, encompassing start lost, stop gained, stop lost, and splice region variants. Approximately 90% of these SNPs were located in the upstream and downstream regions of the CDS region, while the remaining 9% were located within the coding region. There were 41,564 transition and 9297 transversion mutations with a ratio of 4.5. Pairwise SNP variations ranged from 0 to a maximum of 6115 SNPs among any pair of *w*Di strains (Appendix A). It is worth noting that the clustering results, derived from pairwise SNP variations, revealed that *w*DiTW_2 did not cluster with the AS_*w*Di strains; rather, it exhibited a closer association with the NA_*w*Di strains. A phylogenetic analysis was performed utilizing consensus core genome sequences, and the results indicated that the *w*DiTW_2 strain formed an isolated branch (Appendix A), suggesting the possibility of co-infection.

### 2.5. The Conservation of Pathways in wDi Underlines the Wolbachia–Host Symbiotic Relationship

Based on the gene function annotation results obtained through the use of eggNOG-Mapper v2.1.10 [31] against the eggNOG database v5.0.2 [32], KO numbers were assigned to the genes of all 65 *w*Di genomes. Subsequently, we manually identified genes associated with pathways that had previously been recognized as playing a role in symbiosis mechanisms, including thiamine, riboflavin, pyridoxine, biotin, and heme metabolism pathways [3,4,5,34,35]. These pathways were subsequently visualized in a heatmap (Figure 7). The heatmap revealed a high degree of conservation in the majority of pathways across all 65 *w*Di genomes. Specifically, key pathways, including thiamine, riboflavin, and folate biosynthesis metabolism, exhibited conserved patterns similar to those observed in *Wolbachia* genomes from supergroup B [36]. In the pyridoxine metabolism pathway, the presence of *thrC*, which encodes threonine synthase, was absent in the majority of *w*Di genomes. In the fatty acid biosynthesis pathway, a majority of *w*Di genomes lacked the presence of *accA* and *accD*, which encode acetyl-CoA carboxylase carboxyl transferase subunit alpha and beta, respectively. In the purine metabolism pathway, most *w*Di genomes were deficient in *guaA* and *guaB*, which are responsible for encoding GMP synthase and IMP dehydrogenase, respectively. In the pyrimidine metabolism pathway, it appeared to be generally well-conserved in 65 *w*Di genomes, with only a few instances of gene losses. The type I secretion system was consistently conserved across 65 *w*Di genomes. Similarly, the other secretion systems exhibited a high level of conservation, with the exception of the type II secretion system, where *gspD*, responsible for encoding the general secretion pathway protein D, was not present in most NA_*w*Di genomes. In most *w*Di genomes, two complete operons associated with the type IV secretion system (T4SS) were hosted, and the gene *virB6*, which encodes the T4SS protein, was found in high copy numbers. Pathways related to metabolite transport, such as phosphate, lipoprotein, zinc, biotin, iron (III), and phospholipid transport, exhibited a high degree of conservation. In particular, biotin and iron (III) transport were entirely conserved in all 65 *w*Di genomes. It is worth noting that in most NA_*w*Di genomes, there was a notable absence of *CcmC*, which encodes the heme exporter protein C.

### 2.6. Comparative Genomics of NA_wDi and AS_wDi

There were 17 orthogroups encompassing a total of 955 genes uniquely in NA_*w*Di strains, and 5 orthogroups housing 48 genes were identified exclusive to AS_*w*Di strains. A predominant portion of these genes unique to *w*Di strains from Asia or North America coded for ANK proteins (Appendix A). Out of the 955 genes, 451 were subjected to annotation against the COG database, while the 48 genes unique to Asia strains had 9 genes annotated against the same database. Among the 451 annotated genes, associations were found with functional categories such as carbohydrate transport and metabolism; cell wall/membrane/envelope biogenesis; lipid transport and metabolism; cell cycle control; cell division; chromosome partitioning; RNA processing and modification; signal transduction mechanisms; and secondary metabolite biosynthesis, transport, and catabolism. Moreover, within the 955 genes, 55 genes were associated with the ABC transporters (Appendix A). On the other hand, the nine genes unique to Asia were all associated with carbohydrate transport and metabolism.

Using mummer 4.0.0rc1 [38] and *w*DiUY as the query genome, we detected the SNPs and insertions and deletions (indels) between *w*DiUY and GCA_019355355. The results revealed a total of 39,625 SNPs and 2153 indels. Among these SNPs and indels, a detailed breakdown includes 488 frameshift deletions, 482 frameshift insertions, 117 non-frameshift deletions, 120 non-frameshift insertions, 15,433 nonsynonymous mutations, 172 stop gains, 35 stop losses, and 19,406 synonymous mutations. In the SV diagram (Figure 8), we observed four notably shared regions exhibiting reverse matching, indicating inversions between *w*DiUY and GCA_019355355. Furthermore, they also showed 6 inversions, 29 translocations, 65 duplications, 10 SV-based insertions, and 4 SV-based deletions. The 10 SV-based insertions measured 17,680 bp, encompassing a total of 20 genes. Nineteen of these genes were fully covered, and one gene was covered by more than 99%. Notably, these genes were associated with the coding of transposase, phage tail tube protein, group II intron-encoded protein, and hypothetical protein. On the other hand, the four SV-based deletions accounted for 4004 bp, resulting in the complete deletion of five genes in *w*DiUY. These deleted genes were involved in coding transposase and ANK protein.

## 3. Discussion

The earlier comparative analyses and mitochondrial haplotype networks suggest that *D. citri* from Taiwan and Uruguay are more closely related, while California *D. citri* is closely related to Florida *D. citri* [41]. The ANI analysis, PCoA results, and phylogenetic tree support the relationship between geographic populations from the perspective of symbiotic bacteria. In Figure 1, Figure 2, Figure 3 and Appendix A, two distinct clusters representing AS_*w*Di and NA_*w*Di strains are clearly observed. Based on the profiles of the five MLST alleles identified in the 65 *w*Di strains, three distinct STs, ST-173, ST-FL, and ST-462, were identified. ST-173 strains are predominant in China, while ST-FL strains are prevalent in the USA. The *w*DiUY from Uruguay matches ST-173 in the PubMLST database [23]. The *w*DiTW_2 matches ST-462, for which the profiles of the five MLST alleles have not been previously reported. On the other hand, *w*DiTW_1 exhibits a partial match with the ST-173 profile. The partial match with the ST-173 profile, two copies of *coxA* and *fbpA*, and lack of *gatB* and *hcpA* indicate the possibility of genome assembly issues in co-infected populations, which is also supported by the BUSCO score. The resolution of this matter may benefit from the utilization of higher-quality reads. These findings contribute to our understanding of the geographic population structure of *w*Di.

Most of the evidence supporting *Wolbachia* as a nutritional mutualist comes from genomic studies [2,5,42,43]. In the annotation of core genes, a significant number of them are associated with metabolism, including energy, vitamins, nucleotide, and amino acid metabolism (Figure 5). This indicates that *w*Di has the potential to provide nutrients to its host, *D. citri*. The construction of metabolic pathways reveals the conservation of thiamine, riboflavin, and folate biosynthesis metabolism; the type I secretion system; biotin transport; and iron (III) transport pathways in *w*Di (Figure 7). Secretion systems are used by bacterial pathogens to manipulate the host and establish a replicative niche [44]. T4SS is well-documented for its involvement in the infection and survival strategies of a wide array of symbiotic and pathogenic intracellular bacteria [45]. Within the majority of *w*Di genomes, the presence of two complete operons associated with T4SS has been identified (Figure 7). This implies the bacteria’s capability to facilitate the transfer of DNA and/or proteins to eukaryotic cells, potentially playing a crucial role in the manifestation of *Wolbachia*-induced host phenotypes. It is important to consider that due to the incompleteness and high fragmentation of most of the *w*Di genomes analyzed in this study, some genes may be falsely identified as absent in Figure 7.

Within the AS_*w*Di, we selected *w*DiUY as the representative due to its comparatively high genome assembly completeness. Subsequently, we carried out a comparative analysis with NA_*w*Di GCA_019355355. The findings revealed a total of 25 genes associated with SV-based insertions and deletions. These genes include transposase, group II intron-encoded protein, phage tail tube protein, and ANK protein. These insertion sequences and prophages are thought to play pivotal roles in the evolution and adaptation of *W*olbachia [46,47,48,49,50]. Considering that ANK proteins may interact with the host factors in the host cytoplasm, they are presumed to play a significant role in the dynamic interactions between *Wolbachia* and its host [51]. Understanding the distinct functions of genes encompassed within the SV-based insertions and deletions may contribute to our understanding of the differences between *w*Di in North America and Asia.

In this study, the ANI, PCoA, and phylogenetic analyses provide robust evidence that the *D. citri* in Florida did not originate from China. Additionally, our study further classified the 65 *w*Di strains into distinct categories of AS_*w*Di and NA_*w*Di. We conducted a pan-genome analysis of 65 *w*Di genomes, comparing similarities and differences between AS_*w*Di and NA_ *w*Di. Most of the pathways associated with symbiosis mechanisms were found to be conserved, such as thiamine, riboflavin, pyridoxine, biotin, and heme metabolism pathways. In addition, variations involving the insertion or deletion of genes coding for ANK proteins, transposase, group II intron, and phage tail tube proteins were identified in *w*Di strains from Asia and North America. The results obtained from this study have the potential to improve our understanding of *w*Di.

## 4. Materials and Methods

### 4.1. Genome Assembly and Annotation

As of March 2024, the NCBI database currently contains 13 *w*Di genomes, which encompass 4 complete, 3 chromosome, 1 scaffold, and 5 contig-level genomes. In our pursuit to investigate the presence of *Wolbachia* in other *D. citri*, we conducted a screening of available metagenomic data in the SRA database (Appendix A). For this purpose, we downloaded the reads from the SRA database and subsequently carried out an assembly process using Megahit v1.2.9 [52], employing the default parameters. The *w*DiTW_1, *w*DiTW_2, *w*DiCA, and *w*DiUY genomes were generated by using hifiasm-meta v0.3-r063.2 [53] with default parameters. All genomes were reannotated using Prokka v1.14.6 [54], providing files (e.g., faa and gff) for subsequent genomic data analysis. Function annotation of all predicted protein-coding genes of *w*Di was performed by searching against the eggNOG database v5.0.2 [32] using eggNOG-Mapper v2.1.10 [31] with default parameters. Additionally, Pfam domains were annotated using the pfam_scan.pl v1.6 [55] script to search against Pfam database v35.0 [56]. The protein-encoding genes containing ankyrin domains were identified by HMMER v3.3.2 [57] based on the hidden Markov model (HMM) profiles (PF00023, PF12796, PF13606, PF13637, and PF13857) of the ankyrin domains downloaded from the InterPro database (accessed on 15 December 2023, https://www.ebi.ac.uk/interpro/). Specifically, we used the hmmsearch with ankyrin domains to search the ANK proteins, with a threshold of E-value ≤ 1 × 10^−5^, and manually removed redundancy.

### 4.2. Orthogroup Detection, PCoA, and ANI Analysis

Protein-coding sequences from all *w*Di genomes, provided by Prokka v1.14.6 [54], were utilized as input for OrthoFinder v2.5.4 [26] to detect orthologous groups using default parameters. The orthogroup clustering results were transformed into a matrix representing genomes by orthogroups, with the value in each cell corresponding to the copy number. This matrix was then converted into a Jaccard distance matrix among genomes using the VEGAN package v2.6-4 [28] in R. Subsequently, it was processed using the PcoA function in APE v5.7-1 [29] and visualized using ggplot2 v3.4.2 [58]. To further analyze the genetic relationships among the 65 *w*Di genomes, ANI analysis was performed using fastANI v1.33 [25]. The results of the ANI analysis were effectively visualized using pheatmap package v1.0.12 [27] in R.

### 4.3. Pan-Genome Analysis

For each pan-genome or core genome analysis, 65 genomes were randomly sampled without repetition, resulting in 100 possible combinations based on the OrthoFinder v2.5.4 [26] results. To model the pan-genome/core genome, we employed the exponential regression model by fitting medians using the least square method, implemented in the nlsLM function of minpack.lm package v1.2-4 in R [59]. The curve fitting for the pan-genome was executed using the model $y=A_{pan}x^{B_{pan}}+C_{pan}$. In this model, when 0 < *B_pan_* < 1, it indicates an open pan-genome, whereas *B_pan_* < 0 or *B_pan_* > 1 indicates a closed pan-genome. On the other hand, the curve fitting for the core genome was conducted using the model $y=A_{core}e^{xB_{core}}+C_{core}$.

The gene families that are found in all 65 *w*Di strains were categorized as core gene families. Gene families found in 59 to 64 *w*Di strains were designated as softcore gene families. Those present in 2 to 58 *w*Di strains were classified as dispensable gene families. Lastly, gene families found in only one *w*Di strain were identified as private gene families. The frequency and percentage of gene families, as well as the proportion of genes with Pfam domains in core, softcore, dispensable, and private genomes, were effectively visualized using ggplot2 v3.4.2 [58]. For presenting the presence and absence information of pan-genome gene families in the 65 *w*Di genomes, ComplexHeatmap v2.14.0 [37] was employed. To compare gene lengths in core, softcore, dispensable, and private genes, ggplot2 v3.4.2 [58] was also utilized for visualization. To ensure robust statistical analysis, multiple comparisons were conducted using the Student–Newman–Keuls test with a significance level of α = 0.05, facilitated by the agricolae v1.3-6 package [60] in R.

### 4.4. Construction of Metabolic Pathways and Functional Analysis

According to the function annotation results provided by eggNOG-Mapper v2.1.10 [31], the pathways associated with each gene, assigned a KO number, were clearly identified. Several pathways were selected based on prior literature that investigated their significance in the *Wolbachia*–host endosymbiotic relationship. Subsequently, the presence/absence of genes within these selected pathways was manually counted, and the data were visualized as a heatmap using R’s ComplexHeatmap v2.14.0 [37] package. The GO and KO enrichment analysis of core genes was performed using clusterProfiler v4.6.2 [61].

### 4.5. Phylogenomic Analysis

Multiple sequence alignments for single-copy genes were performed using Muscle v5.1 [62] and trimmed using Trimal v1.4.1 [63]. The phylogenetic tree was constructed using IQ-TREE v2.2.2.3 [30] with ultrafast bootstrap mode and 5000 iterations; *A. marginale* (GCA_000020305.1) was used as an outgroup. Branch support was estimated using a Shimodaira–Hasegawa (SH)-like approximate likelihood ratio test with 1000 replicates. Finally, the phylogenetic tree was visualized using ITOL v6 [64].

The identification of core SNPs involved the input of 65 *w*Di genome sequences, with one genome designated as the reference (referred to as GCA_019355355) [65]. This process was carried out using snippy v4.6.0 [33], which utilizes BWA-MEM v0.7.17-r1188 [66] and freebayes v1.3.6 [67] for SNP identification, resulting in the generation of a core genome alignment. To enhance the accuracy of the core genome alignment, Gubbins v 3.3.1 [68] was employed to eliminate recombinant regions, producing a recombination-corrected alignment. Subsequently, snippy v4.6.0 [33] was used to obtain the core SNPs. For the construction of a maximum-likelihood phylogenetic tree from core SNPs with robust statistical support, IQ-TREE v2.2.2.3 [30] was employed, involving 1000 bootstrap replicates. The selected model was K3P + ASC. Finally, the phylogenetic tree was visualized using ITOL v6 [64]. SNPs that impacted annotated portions of the genome were annotated using SnpEff v5.0e [69] to determine the likely effects of variants on gene function.

### 4.6. Identification of Syntenic and Rearranged Regions

Whole-genome alignments were generated using MUMmer v 4.0.0rc1 [38] with default parameters. The 1-to-1 alignment results were then employed to identify SNPs/indels using the delta2vcf module in MUMmer. The many-to-many alignment results were utilized for the identification of SVs through SyRI v1.5.4 [39] and visualized using plotsr v1.1.1 [40]. Subsequent to these analyses, the putative functional effects of the identified SNPs, indels, and SVs were annotated using ANNOVAR [70].

## Figures and Tables

**Figure 1 ijms-25-04851-f001:**
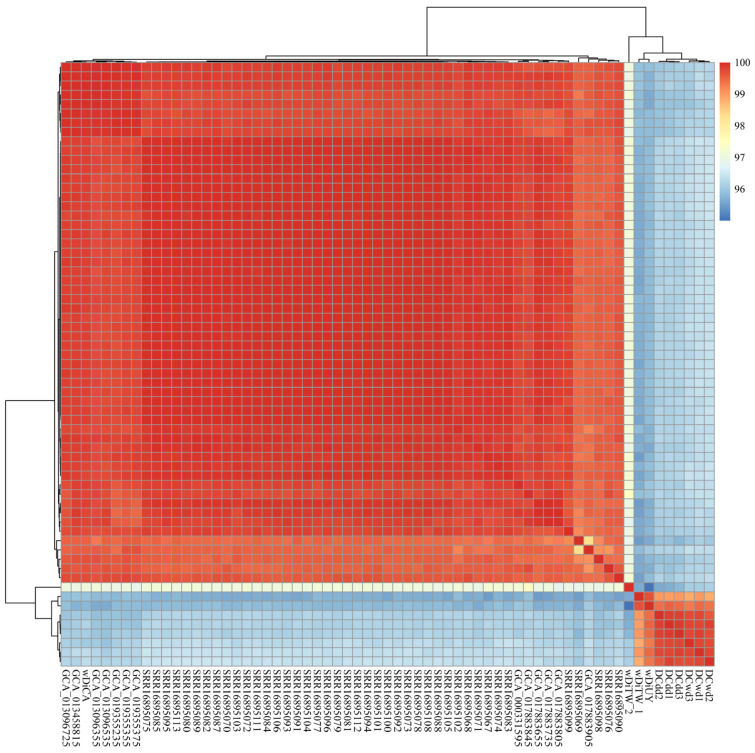
The ANI analysis across all 65 *w*Di conducted using fastANI v1.33 [25] and visualized using pheatmap package v1.0.12 [27].

**Figure 2 ijms-25-04851-f002:**
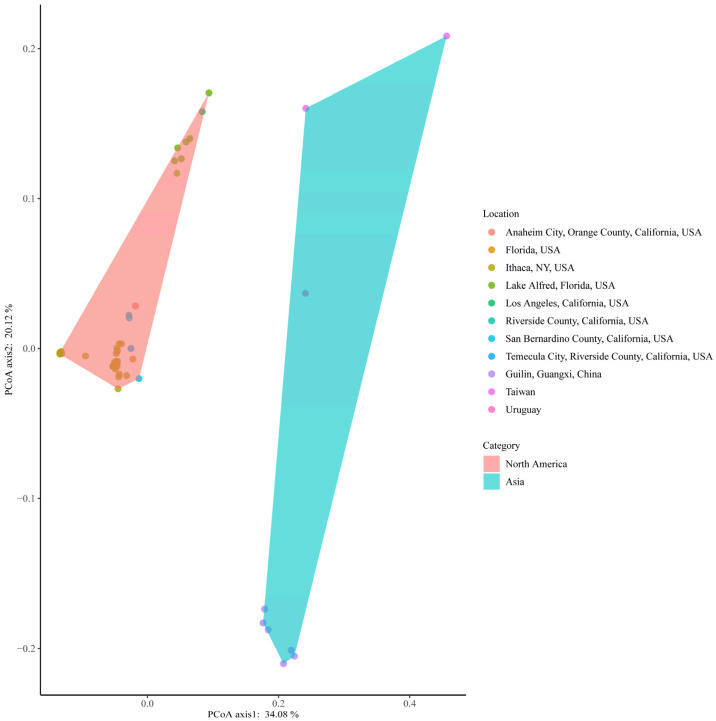
The PCoA of gene copy number conducted using VEGAN package v2.6-4 [28] and ape package v5.7-1 [29].

**Figure 3 ijms-25-04851-f003:**
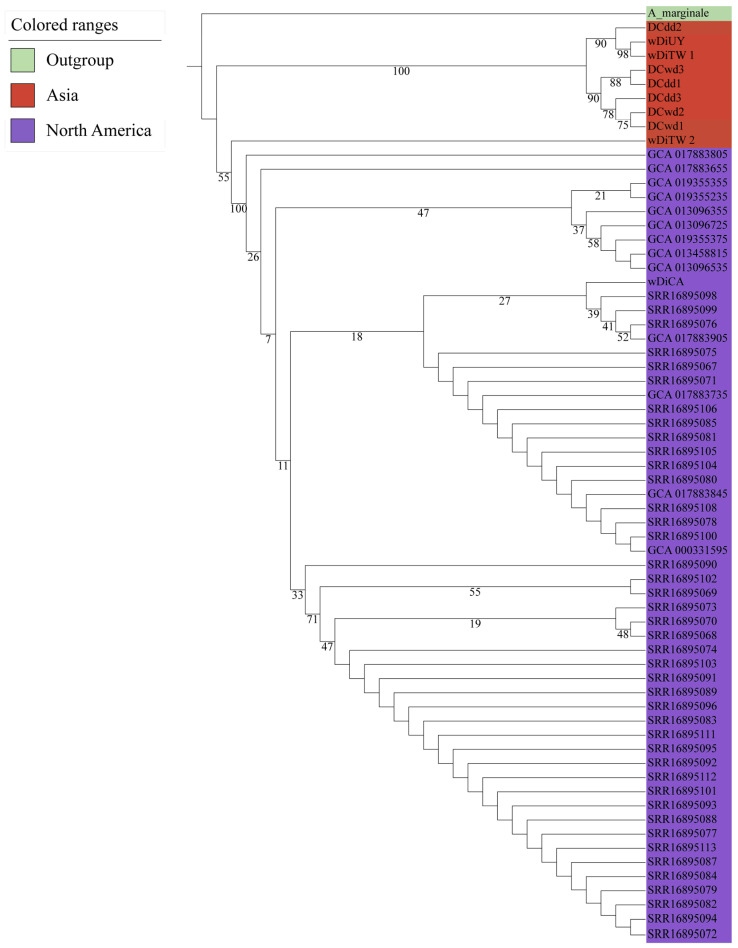
The phylogenetic tree constructed using IQ-TREE v2.2.2.3 [30] with *Anaplasma marginale* set as the outgroup. The amino acid substitution model used was LG + F + G4. Bootstrap values are shown at each node. The *w*Di strains and outgroup are color-coded and shown in colored ranges.

**Figure 4 ijms-25-04851-f004:**
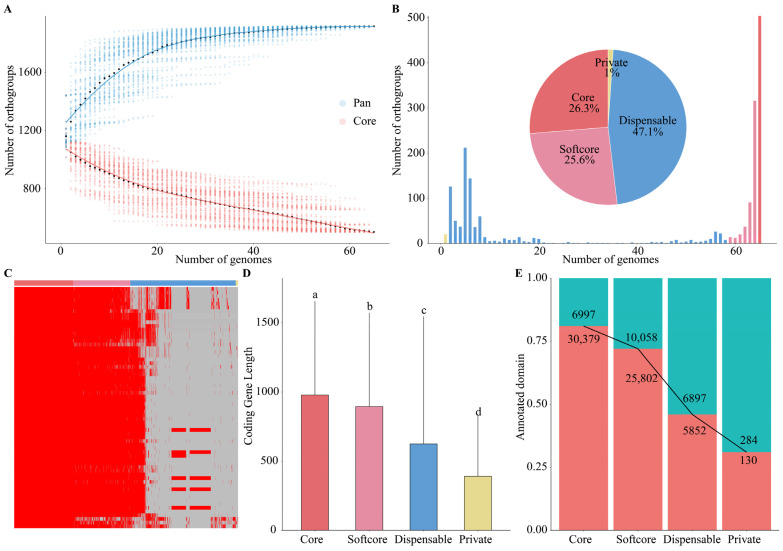
Pan-genome and core genome analyses of 65 *w*Di genomes. (**A**) The core genome and pan-genome development plots. The number of orthogroups in the pan-genome and core genome is shown by blue dots and red dots, respectively. (**B**) Composition of the pan-genome. The histogram shows the number of orthogroups in the 65 genomes with different frequencies. The pie chart shows the proportion of the orthogroup marked by each composition. (**C**) Presence/absence information of orthogroups in the 65 *w*Di genomes. (**D**) Comparison of the length of in core, softcore, dispensable, and private genes. Multiple comparisons are performed using the Student–Newman–Keuls test with a significance level of α = 0.05. Different letters indicate statistical difference in the length of core, softcore, dispensable, and private genes. (**E**) Proportion of genes with Pfam domains in core, softcore, dispensable, and private genes. Red histograms indicate the genes with Pfam domain annotation; green histograms indicate the genes without Pfam domain annotation.

**Figure 5 ijms-25-04851-f005:**
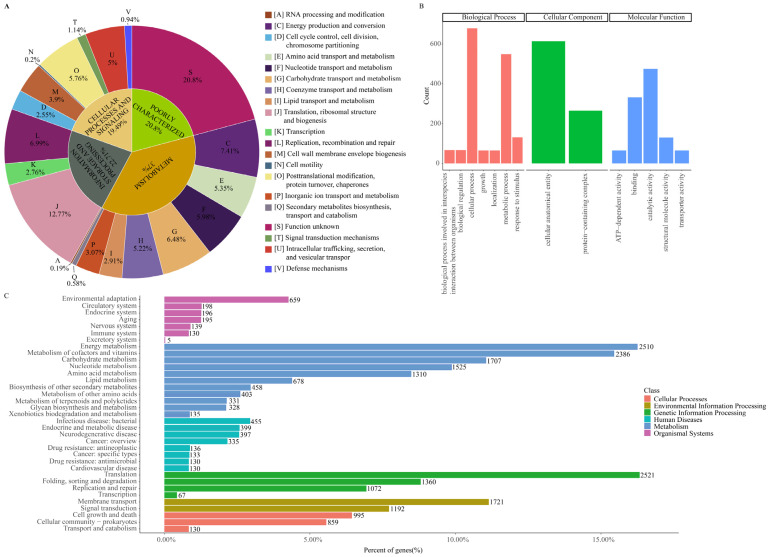
Functional annotation conducted using eggNOG-Mapper v2.1.10 [31] against the eggNOG database v5.0.2 [32]. (**A**) Pie plot displaying the COG function categories of core genes. (**B**) Bar plot illustrating the GO function categories of core genes. (**C**) Bar plot of the KEGG pathway categories of core genes.

**Figure 6 ijms-25-04851-f006:**
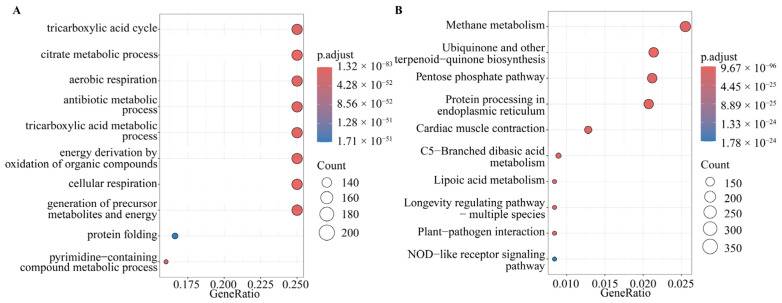
GO and KEGG enrichment analysis of core genes. (**A**) Bubble chart for the GO enrichment analysis of core genes. (**B**) Bubble chart for the KEGG enrichment analysis of core genes.

**Figure 7 ijms-25-04851-f007:**
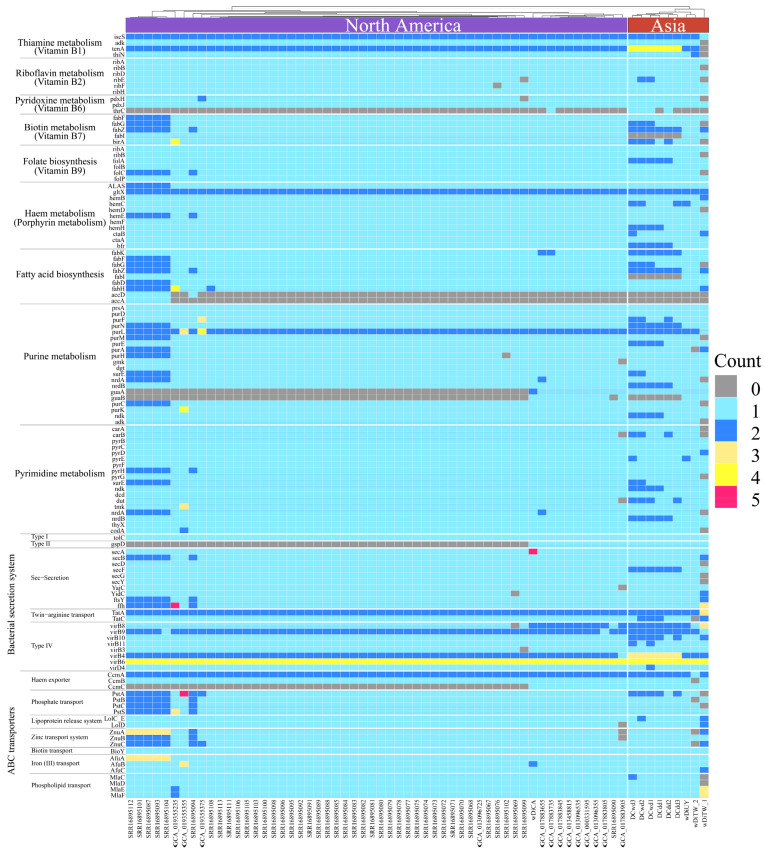
Heatmap of gene presence and absence in metabolic and secretion/transport pathways across 65 *w*Di genomes using Complexheatmap v2.14.0 [37] based on the KO assigned by eggNOG-Mapper v2.1.10 [31]. The genomes analyzed are arranged on the x-axis, with colors representing different *w*Di types. The y-axis represents various genes and metabolic pathways. Different colors in the heatmap indicate the number of genes associated with metabolic and secretion/transport pathways.

**Figure 8 ijms-25-04851-f008:**
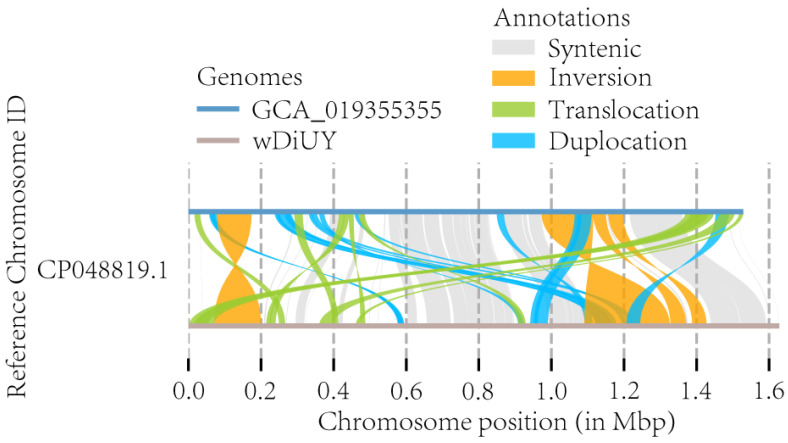
Identifying genome sequence differences between *w*DiUY and GCA_019355355 using SyRI v1.5.4 [39] and visualized using plotsr v1.1.1 [40].

## Data Availability

The 65 *w*Di genome data presented in the study are openly available in FigShare at https://figshare.com/s/516ccbb960c65376b0d2 (accessed on 26 March 2024).

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
