# Peer review of "Pan-Genome Analysis of *Wolbachia*, Endosymbiont of *Diaphorina citri*, Reveals Independent Origin in Asia and North America"

_ijms, 2024, doi:10.3390/ijms25094851_

Round 1

Reviewer 1 Report

Comments and Suggestions for Authors

The MS by Singh et al. describes  a pan-genome analysis of Wolbachia endosymbiont of Diaphorina citri (wDi). Their findings contribute to the understanding of the geographic population structure of wDi and its interaction with host. I would say it is of interest to a wide range of entomologists. However, I have some major concerns that must be addressed before this work can be considered for publication.

First, the MS title says that “the independent evolution” of two wDi strains was revealed. I don’t exactly get it. Do the authors mean that there is active gene transfer within the Asian group, as well as within the North American group? Then it is better to talk simply about the isolation of these two groups. Or do they mean adaptive traits by “evolution”? Then it is necessary to be more specific.

Second, the authors do not provide a phylogenetic tree. It is very unusual for the MS devoted to the pan-genome analysis. The reader need to understand the phylogenetic position of these strains, so reference isolates should be included. The fact that they form separate clusters cannot surprise the reader; isolation by distance is just the reason.

The most of figures are rather insufficient, as the signatures are simply impossible to read.

It is said in the Discussion: «This partial match indicates the possibility of assembly issues in the genome, which is also supported by the BUSCO score.» Does this mean that the authors do not have a complete genome assembly? In this case the claim that they have genomes with two copies of coxA and fbpA, but lost gatB and hcpA, raises questions.

I believe as well that Introduction and Discussion sections could be strongly shortened.

Author Response

Point 1: The MS title says that “the independent evolution” of two wDi strains was revealed. I don’t exactly get it. Do the authors mean that there is active gene transfer within the Asian group, as well as within the North American group? Then it is better to talk simply about the isolation of these two groups. Or do they mean adaptive traits by “evolution”? Then it is necessary to be more specific.

Response 1: Thank you for your comments. The MS title says that “the independent evolution” of two wDi strains was not specific enough. We grouped the 65 wDi genomes into two distinct groups associated with geographic region, Asian groups and North American group, which indicated these two populations likely originated independently. However, we did not conduct an in-depth analysis of the evolution of wDi in this study, such as the whether they had some adaptive traits with the genome evolution. Therefore, we suggest revising “the independent evolution” to “the independent origin” in the MS title.

Point 2: The authors do not provide a phylogenetic tree. It is very unusual for the MS devoted to the pan-genome analysis. The reader need to understand the phylogenetic position of these strains, so reference isolates should be included. The fact that they form separate clusters cannot surprise the reader; isolation by distance is just the reason.

Response 2: Thank you for your comments. We have added phylogenetic tree in the revised manuscript.

Point 3: The most of figures are rather insufficient, as the signatures are simply impossible to read.

Response 3: Thank you for your comments. Most of the pictures are difficult to read, so in the revised manuscript, we have reorganized and adjusted the font size of some images to enhance readability.

Point 4: It is said in the Discussion: «This partial match indicates the possibility of assembly issues in the genome, which is also supported by the BUSCO score.» Does this mean that the authors do not have a complete genome assembly? In this case the claim that they have genomes with two copies of coxA and fbpA, but lost gatB and hcpA, raises questions.

Response 4: Thank you for your comments. The assembled Taiwan wDi genome was 3.2 Mb in length, roughly two times that of the others, which indicated that there were two types of wDi in the Taiwan population. The population of the sequencing Taiwan citrus psyllids were raised in the United States, and it may be a co-infected population or a mixture of two populations that we suspected (Carlson et al., 2022). It is difficult to assemble the genome from a mixed population, as there are many highly similar genomes that make it hard to distinguish them. So there may be some problems with the Wolbachia assembly of Taiwan wDi, resulting in incompleteness in some regions. All other wDi genomes were assembled with no similar problem.

Point 5: The Introduction and Discussion sections could be strongly shortened.

Response 5: Thank you for your comments. We have made the necessary shortening of Introduction and Discussion sections.

Reviewer 2 Report

Comments and Suggestions for Authors

The science seems sound and the conclusion interesting, though I am not an expert in this field.

The last paragraph of the introduction states the conclusions, which I think is not necessary. You already said them in the abstract, so end the introduction with what questions your research answers / hypotheses you wanted to test, and how you did it, without revealing the results. Lines 84-94 can be deleted or moved to the end of the discussion.
By contrast, the last paragraph of the discussion reads like it belongs as the last paragraph of the introduction. I would switch these two and see how that sounds.

The font of Figure 1 is too small to read. I recommend separating these into two figures. Make the ANI figure as large as allowed, such that the font size is equivalent to no smaller than size 8 or 10. For the PCoA, the size of the chart is fine, but change the text to be bigger, ideally size 10 or 12.
In figure 4 and 5, too, you should make the font bigger without enlarging the figure itself, because there is a lot of empty space surrounding the words. You may need photo editing software to do this and not the original graph-making software: delete the original text and add your own in a larger font.
Figure 6 is excellent.

What is the significance of the closed pan-genome, and of the percentage of Pfam domains in core, soft core, dispensable, and private genes? I did not see this explained in the discussion. Also, as the latter four terms are not standardized, but seem to be invented for this study, I recommend defining them in the results rather than just the methods. [You also may want to say something like "semi-core" or "mostly core" instead of "soft core," as "soft core" is commonly used to describe a style of pornography.]

In line 230, is it normal to see SNPs regularly every certain number of bases? Or could that be a computing error?

Comments on the Quality of English Language

A common grammar issue is capitalizing names of proteins or genes or GO terms when this may not be necessary. Please go over the text and double check what is appropriate.

Minor grammar/style fixes:
45 "hosts" without apostrophe
59 "Taiwan in 1907"
78 "of the endosymbiont" or "of endosymbionts"
80 "of D. citri"
90 "wDi. Most"   <- moot if you delete this section

104 delete "Additionally,"
116 delete "they"
119 delete "the"
162 "each. The"
167 "America, and five"

181 "The pie chart shows"
195 Delete "the substantial" as it's redundant with "was particularly high"
195 replace "in the metabolism" with "as metabolism genes"
200 "Among biological"
218 replace "including" with "were"
224 delete "the"
284 "genes uniquely"
285 "orthogroups housing 48 genes were identified exclusive"
295 "Asia all"
297-333 Generally numbers smaller than 10 are written as words: so "four" instead of "4." There are a few such cases here.
299 "SNPs and 2,153"
303 delete "of"

403 & 406 fix the extra spaces before "."
433 the ")" is missing

Author Response

Point 1: The last paragraph of the introduction states the conclusions, which I think is not necessary. You already said them in the abstract, so end the introduction with what questions your research answers / hypotheses you wanted to test, and how you did it, without revealing the results. Lines 84-94 can be deleted or moved to the end of the discussion. By contrast, the last paragraph of the discussion reads like it belongs as the last paragraph of the introduction. I would switch these two and see how that sounds.

Response 1: Thank you for your comments. Switching the last paragraph of the discussion with the last paragraph of the introduction was a good idea, and we have made this adjustment in the manuscript.

Point 2: The font of Figure 1 is too small to read. I recommend separating these into two figures. Make the ANI figure as large as allowed, such that the font size is equivalent to no smaller than size 8 or 10. For the PCoA, the size of the chart is fine, but change the text to be bigger, ideally size 10 or 12. In figure 4 and 5, too, you should make the font bigger without enlarging the figure itself, because there is a lot of empty space surrounding the words. You may need photo editing software to do this and not the original graph-making software: delete the original text and add your own in a larger font.

Response 2: Thank you for your comments. We have divided Figure 1 into two separate figures and adjusted the font size of most figures to enhance readability.

Point 3: What is the significance of the closed pan-genome, and of the percentage of Pfam domains in core, soft core, dispensable, and private genes? I did not see this explained in the discussion. Also, as the latter four terms are not standardized, but seem to be invented for this study, I recommend defining them in the results rather than just the methods. [You also may want to say something like "semi-core" or "mostly core" instead of "soft core," as "soft core" is commonly used to describe a style of pornography.]

Response 3: Thank you for your comments. In the pan-genome analysis, "open" or "closed" serves as descriptive terms for the pan-genome. "Closed" indicates that more genomes were added, and the number of genes in the pan-genome approached saturation, which we pointed out in the results sections. The percentages of Pfam domains in core, soft core, dispensable, and private genes are merely descriptive results. Therefore, we provided only a brief description of these results in the results section and did not explain them in the discussion. The terms "core", "softcore", "dispensable", and "private" are commonly used in pan-genome analysis and are not invented for this study. In the results, we have made the necessary changes to express the meaning of the four terms more clearly.

For the four terms, the references are as follows:
Lian, Q.; Huettel, B.; Walkemeier, B.; Mayjonade, B.; Lopez-Roques, C.; Gil, L.; Roux, F.; Schneeberger, K.; Mercier, R. A Pan-Genome of 69 Arabidopsis Thaliana Accessions Reveals a Conserved Genome Structure throughout the Global Species Range. Nat Genet 2024, 1–10, doi:10.1038/s41588-024-01715-9.

Shi, T.; Zhang, X.; Hou, Y.; Jia, C.; Dan, X.; Zhang, Y.; Jiang, Y.; Lai, Q.; Feng, J.; Feng, J.; et al. The Super-Pangenome of Populus Unveils Genomic Facets for Its Adaptation and Diversification in Widespread Forest Trees. Molecular Plant 2024, 0, doi:10.1016/j.molp.2024.03.009.

Point 4: In line 230, is it normal to see SNPs regularly every certain number of bases? Or could that be a computing error?

Response 4: Thank you for your comments. The certain number of bases is mean that the number of variants per genomic length (number of variants / genome effeective length). In our analysis ,the number of variants was 7,675, and the genome effective length was 1,528,786 bp, meaning that one core SNP was present on average every about 199 bases. This is not a computing error. We make appropriate changes in the revised manuscript.

Point 5: A common grammar issue is capitalizing names of proteins or genes or GO terms when this may not be necessary. Please go over the text and double check what is appropriate.

Response 5: Thank you for your comments. We have carefully reviewed the full text and made necessary corrections to the capitalization problems of the protein, gene and GO term names, in addition to the grammar/style had fixed.

Round 2

Reviewer 1 Report

Comments and Suggestions for Authors

I am OK with the corrections the authors made in the MS and their reply to my comments. The only thing that still can be improved is Figure 5. Its captions are still too small to be easily read. The Journal format allows to make a figure wider (see Figure 2 in IJMS template), I believe it would be better to do so with Figure 5 and perhaps some others.

Author Response

Point 1: The only thing that still can be improved is Figure 5. Its captions are still too small to be easily read. The Journal format allows to make a figure wider (see Figure 2 in IJMS template), I believe it would be better to do so with Figure 5 and perhaps some others.

Response 1: Thank you for your comments. We have resized Figure 5 and several other figures to enhance their readability.